# Novel Insights on the Toxicity of Phycotoxins on the Gut through the Targeting of Enteric Glial Cells

**DOI:** 10.3390/md17070429

**Published:** 2019-07-23

**Authors:** Océane Reale, Antoine Huguet, Valérie Fessard

**Affiliations:** ANSES, Fougères Laboratory, Toxicology of Contaminants Unit, French Agency for Food, Environmental and Occupational Health & Safety, 35302 Fougères, France

**Keywords:** enteric glial cells, phycotoxins, toxicity, high content analysis

## Abstract

In vitro and in vivo studies have shown that phycotoxins can impact intestinal epithelial cells and can cross the intestinal barrier to some extent. Therefore, phycotoxins can reach cells underlying the epithelium, such as enteric glial cells (EGCs), which are involved in gut homeostasis, motility, and barrier integrity. This study compared the toxicological effects of pectenotoxin-2 (PTX2), yessotoxin (YTX), okadaic acid (OA), azaspiracid-1 (AZA1), 13-desmethyl-spirolide C (SPX), and palytoxin (PlTX) on the rat EGC cell line CRL2690. Cell viability, morphology, oxidative stress, inflammation, cell cycle, and specific glial markers were evaluated using RT-qPCR and high content analysis (HCA) approaches. PTX2, YTX, OA, AZA1, and PlTX induced neurite alterations, oxidative stress, cell cycle disturbance, and increase of specific EGC markers. An inflammatory response for YTX, OA, and AZA1 was suggested by the nuclear translocation of NF-κB. Caspase-3-dependent apoptosis and induction of DNA double strand breaks (γH2AX) were also observed with PTX2, YTX, OA, and AZA1. These findings suggest that PTX2, YTX, OA, AZA1, and PlTX may affect intestinal barrier integrity through alterations of the human enteric glial system. Our results provide novel insight into the toxicological effects of phycotoxins on the gut.

## 1. Introduction

Phycotoxins, mainly produced by dinoflagellates, can accumulate in filter-feeding bivalves and provoke human intoxications with clinical symptoms ranging from intestinal to neurological effects [1]. On the basis of their physico–chemical properties and their biological effects, phycotoxins are classified into different groups. Despite the fact that no human fatalities have been reported with lipophilic toxins, the recurrent presence of both regulated and non-regulated lipophilic phycotoxins (including emerging analogs) in shellfish remains a global concern for human health [2]. Among these phycotoxins, pectenotoxin-2 (PTX2) and yessotoxin (YTX) have never been proven to be involved in human intoxications, although toxic effects have been documented in rodents after oral exposure [3,4]. Okadaic acid (OA) is known to induce nausea, diarrhea, vomiting, and abdominal pain in humans [5]. Azaspiracid-1 (AZA1) provokes similar symptoms with nausea, vomiting, diarrhea, and stomach cramps [6]. No human intoxication has been described so far from the cyclic imine 13-desmethyl spirolide C (SPX) [1], although significant concentrations are regularly detected in shellfish [7]. Recently, analogs of palytoxin (PlTX), an amphiphilic phycotoxin, have been detected in marine organisms of the Mediterranean Sea [8] without any report of human intoxication through seafood consumption in this area. However, PlTX has been involved in human intoxications in tropical and sub-tropical areas inducing both neurological and gastro-intestinal symptoms with sometimes human fatalities due to myocardial injury [8]. All these toxins have been shown to act through rather distinct targets. PTX2 inhibits actin polymerization [9,10]; OA is a potent serine/threonine protein-phosphatase inhibitor, which therefore may affect numerous cellular processes [11,12]; SPX is known as an antagonist of muscarinic acetylcholine receptors [13]; and PlTX is a potent inhibitor of the Na^+^/K^+^ ATPase [14,15,16]. On the contrary, the targets for YTX and AZA-1 have not been clearly elucidated [17]. 

Since ingestion represents the main route of human exposure to phycotoxins, the intestine, one of the first organs in contact with food contaminants, can be greatly affected, as underlined by the gastro–intestinal symptoms observed in humans. Oral exposure of rodents has been associated with intestinal damage such as intestinal microvilli disorder [18], dilatation of the small intestine [19], and cell detachment with the separation of the lamina propria from the epithelium [20]. Moreover, in vitro studies have demonstrated the toxicity of lipophilic phycotoxins on human intestinal epithelial cells (IECs) [21,22,23]. 

The intestinal epithelial barrier (IEB) is a highly dynamic and specialized system composed of several cell types such as enterocytes, goblet cells, endocrine cells, and M cells. Recent scientific advances have pointed out that other cells, present in the mucosa, can interact with IECs. Among these cells, enteric glial cells (EGCs), belonging to the enteric nervous system, are found from the ileum to the colon [24]. Due to their close proximity [25,26], physiological crosstalk between EGCs and IECs has been shown to be important for the regulation of homeostasis, gut functions, and intestinal barrier permeability [27]. In vivo rodent models with EGC deletion showed a clear disruption of the IEB [28] with intestinal inflammation [29], persistent diarrhea [30], and loss of barrier integrity [31,32]. EGCs control IEC functions [33] through the release of mediators such as glial cell-derived neurotrophic factor (GDNF), brain cell-derived neurotrophic factor (BDNF), S-nitrosoglutathione (GSNO), and 15dPJGP prostaglandin derivate [34,35]. 

The ability of some phycotoxins to cross the IEB and/or alter its integrity [23,36,37] suggests that cells beneath the epithelium, including EGCs, could be exposed to these toxins. Data in the literature demonstrated that an AZA1 oral treatment of mice induced a peristalsis arrest [19], suggesting alteration of the enteric nervous system. In vitro, the permeability of an IEC monolayer was affected by OA through the secretion of neuropeptide Y by neuroblastoma cells [38]. However, to date no study has investigated if phycotoxins can directly impact the enteric nervous system, and particularly EGCs. 

In this context, we aimed to evaluate and compare the cellular responses of six phycotoxins (PTX2, YTX, OA, AZA1, SPX, and PlTX) on the rat EGC cell line CRL2690. Neunlist et al. have shown that there are no interspecies differences between rat and human EGCs and that similar interactions with human IECs were observed [26]. Moreover, it has been underlined that the EGC2690 cell line is close to primary rat glial cells regarding glial key markers [39]. We therefore investigated cell viability, morphological cell changes, oxidative stress, inflammation, cell cycle, and specific glial markers including glial fibrillary acidic protein (GFAP), calcium-binding protein (S100β), and nitric oxide synthase (iNOS) using RT-qPCR and high content analysis (HCA) approaches.

## 2. Results

### 2.1. Cytotoxic and Morphological Effects of Phycotoxins 

After a 24 h treatment of EGCs, a dose response curve was obtained with four toxins (Figure 1), and IC_50_ were determined as follow: YTX = 14.5 ± 11.1 nM, OA = 75.9 ± 9.2 nM, AZA1 = 7.0 ± 7.5 nM, and PlTX = 0.4 ± 0.1 nM. No significant cytotoxicity was observed following treatment with PTX2 and SPX up to 64 nM and 127 nM, respectively. 

Morphological modifications were also observed (Figure 2). The EGCs from the vehicle control exhibited neurite branching characterized by a broad network. Different morphological modifications were induced depending on the toxin. Although no effect was detected with the neutral red uptake (NRU) assay, PTX2 induced neurite atrophy from 16 nM and was the only toxin where rapid effects on morphological modifications were observed (at 3 h; data not shown). From 10 nM YTX exposure, neurite alterations were observed, with cells presenting few neurites at the highest concentration. From 32.5 nM, OA induced a concentration-dependent cell elongation as well as cell rounding. Neurite atrophy and cell shrinkage were observed with 2.4 nM AZA1 and above. No morphological effects were noticed with SPX treatment up to 127 nM. Above 0.3 nM PlTX exposure, cells depicted an irregular shape combined with a blebbing of the body cell and a loss of neurites.

### 2.2. Intermediate Filament GFAP and Cell Body Area

After 24 h of treatment, a significant modification of GFAP levels was observed with all the toxins except SPX. A concentration dependent increase in GFAP was induced by PTX2, YTX, and AZA1 reaching 2.3-fold, 1.7-fold, and 1.8-fold, respectively, at the highest concentration (Figure 3). OA induced a significant increase in GFAP levels (1.6-fold) but only at the highest concentration. PlTX induced a decrease of GFAP levels (−38%) at the highest concentration (0.5 nM). PTX2 (from 4 nM) provoked cell shrinkage and then significantly reduced cell body area of EGCs up to −40% at 64 nM. YTX exposure showed a slight but non-significant reduction of the cell body area (−12% at 61 nM). OA did not induce any modification of the cell body area. AZA1 also induced a decrease of the cell body area from 2.4 nM, reaching −22% at 19.3 nM. On the contrary, PlTX induced a +38% increase of the cell body area at 0.5 nM that is in accordance with the depicted blebbing of the cell body. No alteration of GFAP amounts and cell body area were observed with SPX treatment. 

### 2.3. Cell Cycle Analysis 

The cell cycle of EGCs was modified following 24 h treatment with 5 out of the 6 toxins (Figure 4). However, with the exception of PTX2, the modifications were not statistically significant. Following treatment with PTX2, YTX, OA, and AZA1, the subG1 phase was 2.2- to 7.2-fold higher than the control depending on the toxin. PTX2 increased the proportion of both G2/M and polyploid cells concomitantly with a decrease in G0/G1 cells. At the highest concentration of YTX, a slight reduction in the number of G2/M cells and an increase of the number of cells in S and G0/G1 phases were observed. AZA1 exposure induced a reduction in the percentage of cells in S and G2/M phases. PlTX and SPX did not induce any significant modification of the cell cycle progression, except a slight decrease of cells in S phase for SPX at the highest dose.

### 2.4. Apoptosis and Genotoxicity 

A concentration-dependent increase of active caspase-3 was observed for PTX2, YTX, and AZA1 (Figure 5). The maximum increase (between 1.6- and 1.8-fold) was similar for the 3 toxins but corresponded also to a 50% decrease in cell count compared to the vehicle control (Figure 5). OA exposure significantly increased active caspase-3 level amounts only at the highest concentration. SPX and PlTX did not affect the amount of caspase-3. The amount of γH2AX significantly increased at 16 nM PTX2 (1.3-fold) reaching 1.5-fold at 64 nM. No effect on γH2AX levels was observed with the other toxins.

### 2.5. NF-κB Nuclear Translocation

No effect on NF-κB nuclear translocation was shown following 3 h treatment with YTX, OA, AZA1, SPX, and PlTX (Figure 6A). With this short time treatment, no diminution of cell count was observed except with PlTX (50% decrease with 2 nM). However, for longer treatment times (8 h), a significant increase of NF-κB nuclear translocation was observed: up to 2-fold for YTX, 4-fold for OA, and 2.5-fold for AZA1 at the highest tested concentration. If no decrease of cell count was observed at 8 h for YTX and AZA1, a marked decrease was noticed following OA exposure. 

For PTX2, nuclear translocation of NF-κB was ambiguous as rapid cell rounding was suspected resulting in quantification artefacts. Therefore, we further investigated the translocation of NF-κB at 3 h using confocal fluorescence microscopy. Contrary to the positive control (50 ng/mL TNFα) that clearly showed nuclear translocation, no difference between PTX2 up to 64 nM and the vehicle control was observed (Figure 6B).

### 2.6. Oxidative Stress

At 4 h treatment, none of the toxins induced oxidative stress except PlTX for which the increase (4-fold at 1 nM) was correlated with a decrease in cell count (Figure 7). At 24 h treatment, PTX2, YTX, and OA increased oxidative stress up to 3-, 1.8-, and 2-fold, respectively, at the highest concentration. A slight but non-significant increase was also noticed for AZA1 at 19.5 nM. Overall, oxidative stress was correlated to a decrease of cell count. SPX exposure up to 127 nM did not induce oxidative stress in EGCs whatever the time and concentration of exposure. 

### 2.7. S100β and iNOS Production

After 24 h treatment, a significant increase of S100β amount was observed with PTX2, YTX, OA, and AZA1 reaching 1.5-fold to 2.1-fold depending on the toxin and concomitantly with a decrease of cell count (Figure 8). No effect was detected with SPX and PlTX, despite an important decrease of cell count for PlTX at the highest concentration. The responses for iNOS amount were very similar to S100β for PTX2, YTX, OA, and AZA1 with increases reaching 1.5-fold to 2.2-fold depending on the toxin (Figure 9). iNOS levels were not affected by SPX and PlTX.

### 2.8. Modulation of Gene Expression Following Treatment with PTX2, YTX, and AZA1

We selected the three toxins, PTX2, YTX, and AZA1, for additional investigation on the expression of key genes involved in viability, morphology, cell cycle, inflammation, oxidative stress, and gliomediators. Table 1 presents only the statistically significant results and the whole dataset is included as Appendix A. An up-regulation of genes associated with cell mortality was observed with a 1.6-fold increase of BNIP3 at 2 nM YTX and 1.5 nM AZA1, and 1.6-fold increase of GABARAP at 1.5 nM AZA1. CASP3 was significantly decreased with PTX2 at the highest dose (−52% at 4 nM). FOS, a gene involved in apoptotic response, was increased at the highest concentration for YTX and AZA1. Complementary to FOS, BCL2 was decreased for YTX at the highest concentration. The expression of cell cycle genes was significantly affected by two toxins. YTX induced decreases in CDK1(−78%), CDK2 (−42%), and CCNA2 (−77%) expression at the highest concentration. Similarly, AZA1 (at 1.5 nM) induced a down regulation of CDK1 (−66%), CDK2 (−27%), and CCNA2 (−57%). PTX2 induced a trend of decrease for CDK1 gene expression at 4 nM. Modification of the expression of genes involved in inflammation was observed with the 3 toxins. PTX2 significantly induced an up regulation of 1.4-fold of MAPK8 and MYD88 expression, respectively, at 2 nM and 4 nM. YTX induced a significant increase of chemokine CCL2 (up to 2.2-fold at 2 nM) as well as significant down regulations of receptor IL1R1 (−53% at 4 nM), RHOA (−22% at 2 nM), and FZD4, an anti-inflammatory signaling receptor, (−59%). AZA1 induced a significant increase of MAPK8 (1.5-fold at 0.38 nM), CCL2 (up to 2.5-fold at 1.5 nM), and TLR4 (1.8-fold at 1.5 nM). For genes related to oxidative stress, PTX2 induced a decrease of CAT expression (−25% at 4 nM), while AZA1 induced a significant increase of NFE2L2 (1.5-fold at 1.5 nM). Exposure to PTX2 resulted in the up regulation of gliomediator genes: BDNF and GNDF expression increased up to 5.2 and 2.7-fold, respectively, at the highest concentration, although this was not statistically significant. YTX induced a significant 1.5-fold increase of GDNF. For channel and glioreceptor genes, our results showed a 50% decrease of GJA1 with 4 nM YTX. GFRA1 expression, a receptor of GDNF, was 1.4-fold increased with 2 nM PTX2, 1.3-fold with 4 nM YTX, and 1.4-fold with 0.38 nM AZA1. LPA1, a gene coding for adrenergic receptor, was increased up to 1.7-fold with 4 nM PTX2.

## 3. Discussion

Our results show that PTX2, YTX, OA, AZA1, and PlTX cause morphological alterations of EGCs, particularly through disturbance of the neurite network, suggesting that communication between glial cells and other cell types may be affected. Indeed, the network of neurites is highly involved in several interactions, such as binding and transmission of mediators [33,40]. Similar to the morphological alterations observed in EGCs, PTX2, YTX, OA, AZA1, and PlTX have previously been shown to disrupt the cytoskeleton in various mammalian cells through the disturbance of F-actin network in different cell types [38,41,42,43,44,45,46]. As observed in our study, AZA1 was previously shown to decrease the number of neurites in BE(2)-M17 cells [47] and induced an irreversible neurite rearrangement in mouse primary neurons after 48 h of treatment [48]. Similarly to our result, no cytoskeletal alterations have been observed with SPX on various cell lines [49]. Among the five toxins, only PTX2 induced a decrease of cell body area in EGCs without any effect on cell viability. Similar observations were reported with PTX2 on an epithelial rat liver cell line (Clone 9) at 24 h treatment [50]. The morphological changes of EGCs were consistent with variations in GFAP levels, an intermediate filament protein as a key marker of glial cell morphology. As a hallmark feature of glial stress, GFAP level is increased by cell injuries and its level correlates with the functional state of EGCs [31,51]. A decrease of GFAP intensity was observed with PlTX, which could be a consequence of the increase of body cell area through blebbing as the increase of cellular area would lead to a decrease of the GFAP intensity per cm^2^. 

No significant modifications were observed with PTX2, YTX, and AZA1 for calpain genes (CAPN1 and CAPN2), a protease involved in GFAP degradation, which reinforces the hypothesis that GFAP was not adversely affected. Interestingly, gene expression analysis indicated no significant modifications for GFAP except a slight decrease with PTX2. However, the discrepancy between gene expression and protein staining could be related to the slow turn-over of GFAP in glial cells (in vitro half-life of 3–4 days) [52,53], which may explain why we did not observe any decrease of GFAP protein with PTX2. It is noteworthy that a GFAP increase was observed in inflamed regions of the intestine [51,54,55] and that cytokine stimulation could control GFAP up-regulation in enteric glial cells [51].

In agreement with our results, intestinal inflammation has been induced by various phycotoxins: presence of infiltrated lymphocytes detected in the gut submucosa of mice after YTX exposure [56], the presence of neutrophils in the lamina propria [19] as well as infiltration of inflammatory cells in lung and liver [20] described with AZA1 and induction of an inflammatory response with OA and AZA1 in various cell lines [22,57,58]. 

Besides, NF-κB nuclear translocation, a transcription factor involved in the inflammatory response [59], as well as expression of inflammation genes were induced in EGCs with YTX and AZA1. As reported by Nezami et al., abdominal pain is a manifestation of inflammation of the enteric nervous system due to the secretion of inflammatory cytokines and mediators [60]. Therefore, the pain observed after OA and AZA1 intoxication could be related to an inflammatory response. Although PTX2 did not affect NF-κB translocation as previously reported [61], a significant increase in the expression of genes involved in inflammatory processes were observed, suggesting that PTX2 might induce inflammation through another pathway than NF-κB.

In addition, our data showed that toxins can alter key functions of EGCs as both S100β and iNOS levels were increased concomitantly after OA, PTX2, YTX, and AZA1 exposure. S100β, which is induced by activation of the RAGE receptor and MyD88 protein [31], was described to enhance iNOS expression [62,63] and lead to inflammation through the NF-κB pathway [62,64,65], induction of cytoskeletal modifications [66,67], and apoptosis [68]. Up regulation of S100β was described as an adaptive response to the stress of EGCs [63]. iNOS also acts as an inflammation mediator [69,70,71]. Furthermore, the production of iNOS contributes to glial dysfunctions, increasing oxidative stress [72,73,74]. While our study is the first one exploring S100β and iNOS responses after phycotoxin exposure, Franchini et al. previously observed an increase of S100 protein in Purkinje cells after an intraperitoneal injection of YTX [75]. 

Our data revealed that PTX2, YTX, OA, and AZA1 induced apoptosis in EGCs supported by the activation of caspase-3 and the increase of subG1 cells, particularly for PTX2. Caspase-3 activation has been described in other mammalian cells with PTX2 [76], YTX [77,78], OA [47,79,80], and AZA1 [81,82]. Nevertheless, the level of γH2AX, a marker of DNA double strand breaks, was induced only with PTX2. Shin et al. (2011) have previously shown that PTX2 induced apoptosis and DNA breaks following actin disruption in MCF-7 cells [83]. We suggest that apoptosis occurred later with YTX, OA, and AZA1, or that DNA repair was more efficient. A slight decrease of cells in G2/M combined with a slight increase of cells in G0/G1 was induced with YTX that could be a reflection that cells were entering into a repairing state. Although DNA repair involvement has been shown with OA [84] and YTX [85], further investigation is needed to explain these differences.

In addition to a subG1 increase, PTX2 induced an arrest of EGCs in G2/M and an increase of polyploid cells. The G2/M blockage by PTX2 has been reported with other mammalian cells [86,87] and it has been proposed to be a consequence of the disturbance of actin polymerization [88]. Frizzo et al. have reported that a large amount of S100β can inhibit microtubule elongation, causing a rapid disassembly and cell cycle perturbations [67]. Therefore, the increased levels of S100β observed with PTX2 could be linked to G2/M arrest. 

Previous studies have shown that the production of iNOS induces oxidative stress by NO production [72,73,74]. NO production results in apoptosis, mitochondrial respiration failure, and glial degeneration mediated through covalent S-nitrosylation of proteins that affect their activities. The level of oxidative stress was significantly increased in PTX2-, YTX-, OA-, and AZA1-treated EGCs. In agreement with our results, PTX2, YTX, and OA were shown to increase ROS production described in a variety of mammalians cell lines [89,90,91,92,93]. Although AZA1 induced some oxidative stress in EGCs (not statistically-significant), Vale and Hjornevik did not report ROS production in rat PC12 cell line and in primary cultures of cerebellar granule cells with AZA1 [94,95]. To summarize, our observations in EGCs suggest the involvement of the S100β–iNOS-oxidative stress pathway in EGCs following PTX2, YTX, OA, and AZA1 exposure. 

Gliomediators such as GDNF and BDNF produced by EGCs are involved in preserving intact epithelial lining and promoting regeneration of IECs [96,97]. We observed that PTX2 and YTX treatment increased the expression of some gliomediator genes, but this observation should be confirmed at the protein level. As previously described [98,99], the increase of GDNF and BDNF gene expression with PTX2 and YTX seems to be a protective response of EGCs in order to counteract the increased caspase-3, S100β, and iNOS levels [100]. Indeed, GDNF and BDNF are known as important endogenous factors for the regulation of apoptosis in EGC cells [98,99]. Although we did not investigate the effects of OA on the expression of gliomediators, Louzao et al. reported the capacity of OA to induce the neuropeptide Y (NPY) in SH-SY5Y neuroblastoma cells [38]. NPY inhibits electrolyte secretion and is promoted by GDNF in the enteric nervous system [101].

In contrast to these four toxins, PlTX did not show any increase of active caspase-3, γH2AX, S100β, iNOS, nuclear translocation of NF-κB, nor modifications of the cell cycle in EGCs. Previous studies have already pointed out the absence of caspase activation but rather the induction of the cell death in a necrosis-like manner by PlTX in various cells [44]. The rapid cell death of EGCs (beginning at 3 h of exposure) and the increase of cell body area with PlTX are in favor of a necrotic process. Disturbance of ion flux by PlTX [21] provokes a change of osmotic pressure and an increase of the cytoplasmic area leading to necrosis [14,102]. As PlTX did not increase glial specific markers, early oxidative stress may be induced independently of the S100β–iNOS pathway. The early ROS production is probably not cell-type dependent since Pelin et al. have observed the same response in human keratinocytes [103].

Among the six toxins tested in our study, SPX showed a distinct lack of response on EGCs as none of the endpoints investigated was affected by SPX exposure. The absence of cytotoxicity by SPX was also reported using a large range of cell lines [49]. SPX is documented to inhibit both muscarinic (mAChr) and nicotinic acetylcholine receptors (nAChr) [104]. Recent studies showed that EGCs support enteric neurons through the expression of a large number of neurotransmitter receptors, such as mAChr [33,101], especially both M3 and M5 subtypes [105]. Although SPX induced the inhibition of M3 mAChr, it did not alter the level of M3 mAChR protein, and its toxicity seems mainly mediated by nicotinic rather than muscarinic AChR [106,107]. These observations strengthen the absence of effects of SPX on EGCs in our study but did not exclude that effects on enteric neurons could occur and then affect neurotransmission, for example.

We have highlighted that each toxin induced a different response in EGCs, as described in previous studies, evoking singular mechanisms of action [22,108]. Table 2 summarizes the results obtained in EGCs and illustrates the complexity of toxicological responses induced by each toxin. Figure 10 resumes our observations on EGCs for each toxin and highlights which effects are common or specific. PTX2, YTX, OA, and AZA1 globally induced the same effects on EGCs but only PTX2 activated the gliomediators expression. Instead of the other toxins, SPX did not impact EGCs while PlTX induced a rapid cytotoxicity.

## 4. Materials and Methods 

### 4.1. Chemicals 

Penicillin, streptomycin, fetal bovine serum (FBS), and Dulbecco’s Modified Eagle’s Medium with high glucose (DMEM) were purchased from Gibco (Cergy-Pontoise, France). Bovine serum albumin (BSA), Tween 20, Triton X-100, and neutral red were supplied by Sigma-Aldrich (Saint Quentin Fallavier, France). OA, PTX2, YTX, AZA1, and SPX were purchased from the National Research Council Canada (Halifax, NS, Canada) and were dissolved in methanol (MeOH). PlTX was supplied by Wako Chemicals GmbH (Neuss, Germany) and dissolved in ultra-pure sterile water.

### 4.2. Cell Culture and Toxin Exposure

The rat enteric glial cell line (EGC) (CRL2690) was obtained from the American Type Culture Collection (Manasas, VA, USA). Cells were grown in DMEM supplemented with 10% FBS, 50 IU/mL penicillin, and 50 µg/mL streptomycin at 37 °C and 5% CO_2_. Cells were used at passages from 38 to 58. For subculture, cells were seeded in 75 cm^2^ culture flasks and passaged twice a week. For experiments, cells were seeded at 30,000 cells/cm^2^ in 96-well plates for cytotoxicity and HCA assays, and at 40,000 cells/cm^2^ in 12-well plates for qPCR assay. The day after seeding, cells were exposed to toxins in FBS free medium. Two vehicle controls (2.7% sterile water for PlTX, 1.25% MeOH for the other toxins) were included in each experiment.

### 4.3. Cell Morphology and Neutral Red Uptake Assay 

After 24 h of treatment with toxins, cell morphological changes were observed by phase contrast microscopy (Leica Microsystems, Wetzlar, Germany), and the neutral red uptake (NRU) assay was performed as previously described [22]. Absorbance was measured at 540 nm with a microplate reading spectrofluorometer (Fluostar OPTIMA, BMG Labtech, Champigny sur Marne, France). Three independent experiments were performed, and for each experiment, the median of the three technical replicates was expressed relative to that of the vehicle control. When possible, the IC_50_ was determined using GraphPad Prism Software (La Jolla, CA, USA).

### 4.4. High Content Analysis Multiparametric Toxicity Assays Oxidative Stress

After 4 and 24 h of toxin treatment, cells were incubated at 37 °C for 60 min with 10 µM CM-H_2_DCFDA (Thermo Scientific, Waltham, MA, USA) in DMEM, and then for 10 min with 3 µg/mL Hoescht 33,342 (Sigma-Aldrich) in DMEM. Fluorescence was monitored with an Arrayscan VTI HCS Reader (Thermo Scientific) associated with a live cell chamber. The Target Activation module of the BioApplication software was used to quantify the oxidized DCF in the cytoplasm. Cell count was performed using Hoescht labelling. Hydrogen peroxide (H_2_O_2_) (70 µM, Gifrer Barbezat, Decines, France) was used as the positive control. Four independent experiments were performed, and for each experiment, the average intensity of the three technical replicates was expressed relative to that of the vehicle control. 

### 4.5. Cell Cycle Analysis, Inflammation, Genotoxicity, Apoptosis, and Glial Cell Markers

The immunofluorescence detection of the different markers was performed as previously described [109] with the following modifications. Antibodies were purchased from Abcam (Cambridge, UK) and BD Pharmingen Biosciences (Franklin Lakes, NJ, USA), and diluted as follows: mouse anti-GFAP (1/100, BD-556328), rabbit anti-active caspase-3 (1/1000, ab13847), mouse anti-γH2AX ser139 (1/1000, ab26350), rabbit anti-NF-κB-p65 (1/1000, ab16502), rabbit anti-S100β (1/350, ab52642), mouse anti-iNOS (1/350, ab49999), goat anti-mouse IgG H&L Alexa Fluor^®^ 555 (1/1000, ab150114), and goat anti-rabbit IgG (H&L) Alexa Fluor^®^ 647 (1/1000, ab150079). Labeling was performed after 24 h of treatment with the toxins except for NF-kB, for which a shorter time of exposure (3 and 8 h) was chosen.

Cells were classified in the different cell cycle phases through DAPI (Sigma-Aldrich) labelling using the Cell Cycle Analysis module of the BioApplication software. Results were expressed relative to the total cell number. Cytochalasin B (4.5 µg/mL, Sigma-Aldrich) was used as the control for G_2_/M phase calibration. The Target Activation module of the BioApplication software was used to quantify γH2AX (DNA double strand breaks) in the nucleus, and active caspase-3 (apoptosis), iNOS, and S100β (glial cell markers) in the cytoplasm. Methylmethanesulfonate (MMS 400 µM, Acros Organics, Fairlawn, NJ, USA) and cytomix (50 ng/mL TNFα (BD Pharmingen Biosciences) + 50 ng/mL Il1β (Sigma-Aldrich) + 50 ng/mL IFN-γ (Thermo Fisher, Walthan, MA, USA)) were used as positive controls for γH2AX and iNOS, respectively.

NF-κB-p65 nuclear translocation was analyzed using the Compartmental Analysis module. TNFα (50 pg/mL) was used as a positive control. Intermediate filament GFAP was described as a key marker of glial cell functions [63]. The Neurite Detection module was used to quantify both GFAP in the cytoplasm and body cell area.

For each well, 8 fields (20× magnification) were analyzed. Cell counts were performed using nuclear DAPI labelling. Three independent experiments were performed and, for each experiment, the median intensity of three technical replicates was expressed relative to that of the vehicle control. For NF-κB-p65 nuclear translocation, the (nuclear–cytoplasm) difference of intensity was determined and expressed relative to that of the vehicle control.

### 4.6. Confocal Microscopy Imaging 

After 3 h of treatment with 64 nM PTX2, immunostaining of NF-κB-unit p65 was performed as described above. Analysis was performed using an inverted laser-scanning confocal microscope SP8 DMI 6000 CS (Leica Microsystems, Wetzlar, Germany). Images were analyzed with the LAS AF 3.3.0.10134 software (Leica Microsystems, Wetzlar, Germany) and assembled using the ImageJ software.

### 4.7. RT-qPCR

After 24 h of treatment with PTX2 (1, 2, and 4 nM), YTX (1, 2, and 4 nM), and AZA1 (0.38, 0.75, and 1.5 nM) (dose corresponding at the IC_20_), total RNA was isolated, quantified, and assessed for integrity as previously described [110]. A negative extraction control was included for contamination assessment. Reverse transcription (RT) was performed with 1 µg of total RNA using the Transcriptor Universal cDNA Master kit (Roche, Mannheim, Germany) according to the manufacturer’s instructions. Reaction volume was set to 20 µL and RT was performed at 55 °C for 10 min prior to a stopping step for 5 min at 85 °C. Negative RT control of RNase-free water and a no-reverse transcription control (replacement of reverse transcriptase by RNase-free water) were included for assessing, respectively, external contamination and absence of DNA occurring during RNA extraction. The sequences of targeted genes were obtained from the National Center for Biotechnology Information GenBank sequence database (https://www.ncbi.nlm.nih.gov/). For primers, design and in silico analyses for their specificity were performed together using the Primer Basic Local Alignment Search Tool (http://www.ncbi.nlm.nih.gov/tools/primer-blast) with, for each gene, at least one primer designed on an exon–exon junction when possible. All primers were purchased from Sigma-Aldrich. Additional information on target genes and oligonucleotide primers are listed in Appendix A. Quantitative PCR was performed with a LightCycler^®^ 480 (Roche) in 384-well plates. Reactions were carried out on two technical replicates in a total volume of 5 μL containing 1X LightCycler 480 SYBRGreen I Master (Roche), 300 nM of each primer, and 2.5, 0.6, or 0.2 ng cDNA, depending of the target gene. Negative quantitative PCR controls of RNase-free water were included in each run for contamination assessment. The thermal cycling conditions were 95 °C for 5 min, followed by 45 amplification cycles of denaturation at 95 °C for 10 s, annealing at 60 °C for 10 s, and polymerization at 72 °C for 10 s. In order to check the specificity of each amplicon, the melting curve was assessed from 60 °C to 95 °C with a slow temperature ramp (0.06 °C/s) and 10 acquisitions per 1 °C. LightCycler^®^ 480 software was used for quantitative analysis. Calibration curves were established for each gene from a serial two-fold dilution of a reference sample (pool of cDNA samples). According to these calibration curves, for each sample, mean relative amounts of mRNA of the targeted genes were calculated and then normalized to that of the reference gene. Using NormFinder software, the gene GAPDH was chosen as a reference gene since it did not exhibit any significant variation of expression among the samples. The normalized means were used for statistical analyses and values were presented as arbitrary units. Three independent experiments were performed.

### 4.8. Statistics

GraphPad Prism software was used for statistical analyses. An analysis of variance (ANOVA) was performed, and, when the effect of concentration was significant (*P* < 0.05), the values were compared to the control using the Dunnett’s test. Differences were significant at *P* < 0.05. The values presented are means ± SEM.

## 5. Conclusions

In conclusion, our data provide novel insight on the toxicity of phycotoxins on glial cells present along the gastro–intestinal tract. This study demonstrated that EGCs are sensitive to most of the tested phycotoxins, with the exception of SPX. 

Globally, our results suggest that the response of phycotoxins on the intestinal epithelium may involve the enteric nervous system and that EGCs may play a role in the symptoms induced by OA, AZA1, and PlTX in humans. Although no effects have been reported in humans from PTX2 and YTX, these toxins could affect EGCs. Further toxicological investigations are required to elucidate the role of EGCs in the acute toxicity of phycotoxins.

## Figures and Tables

**Figure 1 marinedrugs-17-00429-f001:**
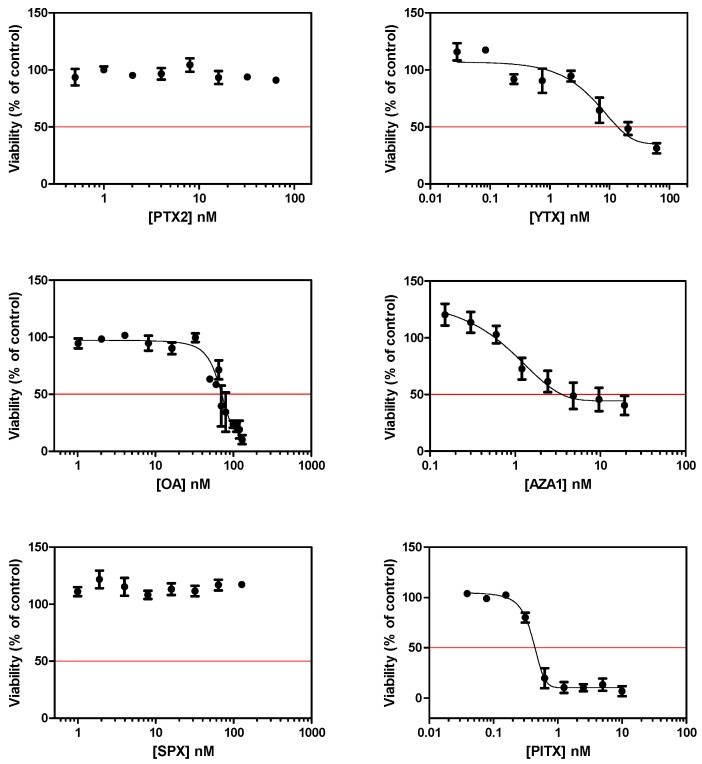
Cytotoxicity on enteric glial cells (EGCs) after a 24 h treatment with pectenotoxin-2(PTX2), yessotoxin (YTX), okadaic acid (OA), azaspiracid-1 (AZA1), 13-desmethyl spirolide C (SPX), and palytoxin (PlTX). Cytotoxicity was measured by the neutral red uptake assay (NRU). Values are presented as mean ± SEM and expressed as percentages relative to the vehicle control medium. Three independent experiments were performed.

**Figure 2 marinedrugs-17-00429-f002:**
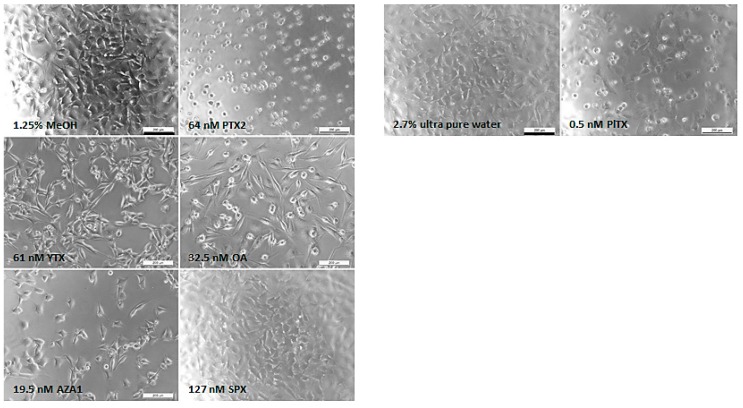
Morphological modifications of EGCs after 24 h exposure to PTX2, YTX, OA, AZA1, SPX, and PlTX. Evaluation of cell morphology was carried out by phase contrast microscopy. Each image is representative of three independent experiments. Vehicle controls: 1.25% MeOH and 2.7% ultra-pure water (for PlTX only). Scale bar = 200 µm.

**Figure 3 marinedrugs-17-00429-f003:**
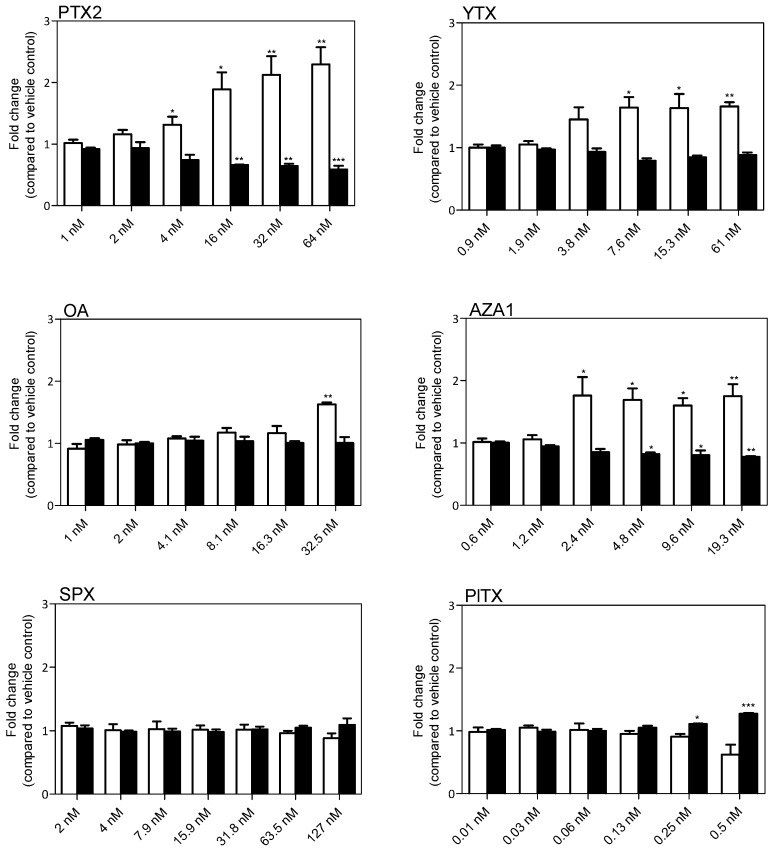
Glial fibrillary acidic protein (GFAP) levels and cell body area in EGCs after 24 h exposure to PTX2, YTX, OA, AZA1, SPX, and PlTX. GFAP levels (white) and cell body area (black) were carried out by Hight Content Analysis (HCA). Values are presented as mean ± SEM and expressed as fold change compared to the vehicle control set to 1. Three independent experiments were performed. *, **: values significantly different from the vehicle control (respectively *P* < 0.05 and *P* < 0.01).

**Figure 4 marinedrugs-17-00429-f004:**
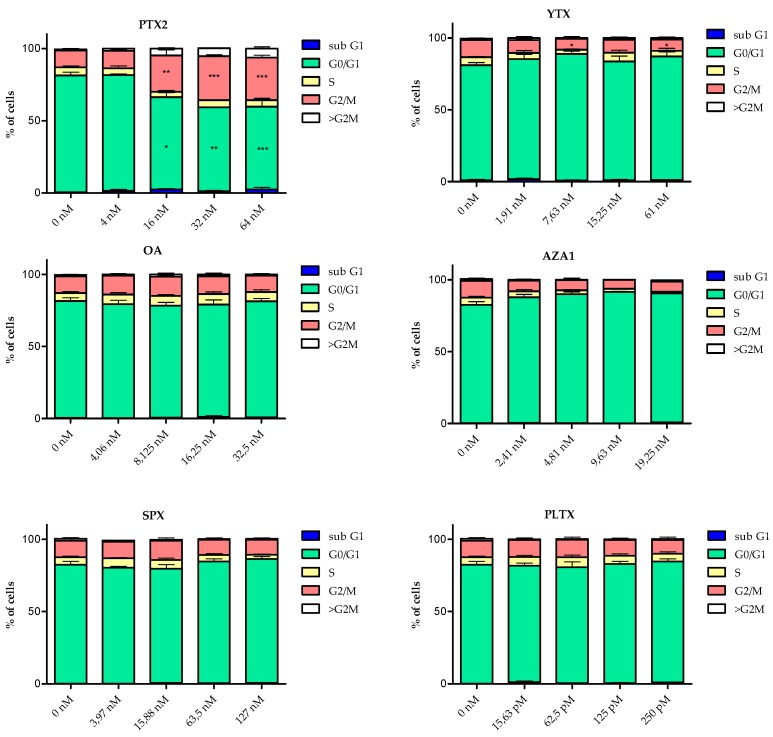
Cell cycle analysis of EGCs after 24 h exposure to PTX2, YTX, OA, AZA1, SPX, and PlTX. The classification of cells in the different cell cycle phases was determined using nuclear DAPI labelling and is expressed relative to the percentage of cells in each phase. Values are presented as mean ± SEM. Three independent experiments were performed. Vehicle controls were 1.25% of MeOH and 2.7% ultra-pure water (for PlTX only). *, **, ***: values significantly different from the vehicle control (respectively *P* < 0.05, *P* < 0.01 and *P* < 0.001).

**Figure 5 marinedrugs-17-00429-f005:**
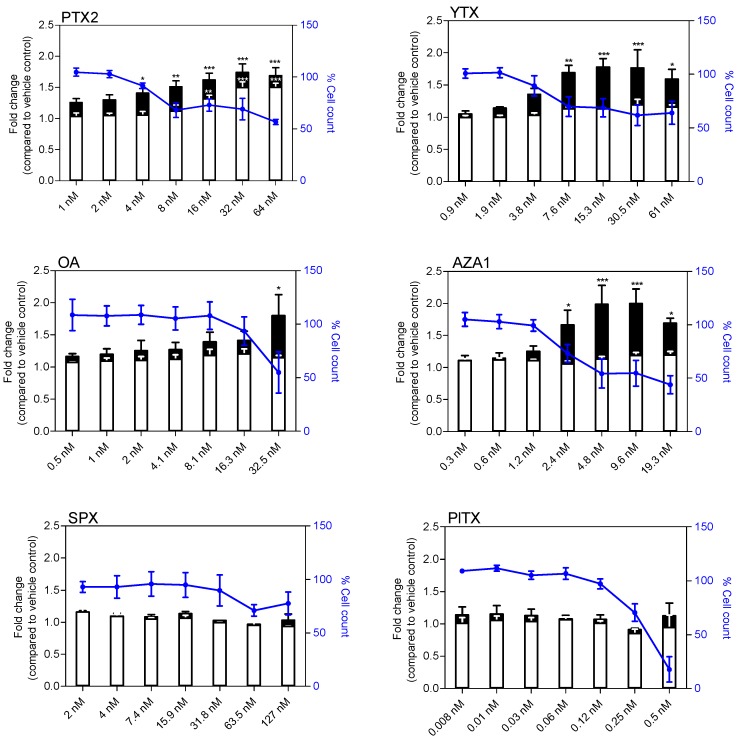
Apoptosis and genotoxicity in EGCs after 24 h exposure to PTX2, YTX, OA, AZA1, SPX, and PlTX. Active caspase-3 (black) and γH2AX (white) were carried out by HCA. DAPI staining was used for cell count (blue). Active caspase-3 and γH2AX are expressed as fold change compared to the vehicle control set to 1. Cell count values are expressed as percentages of the vehicle control. Values are presented as mean ± SEM. Three independent experiments were performed. *, **, ***: values significantly different from the vehicle control (respectively *P* < 0.05, *P* < 0.01, and *P* < 0.001).

**Figure 6 marinedrugs-17-00429-f006:**
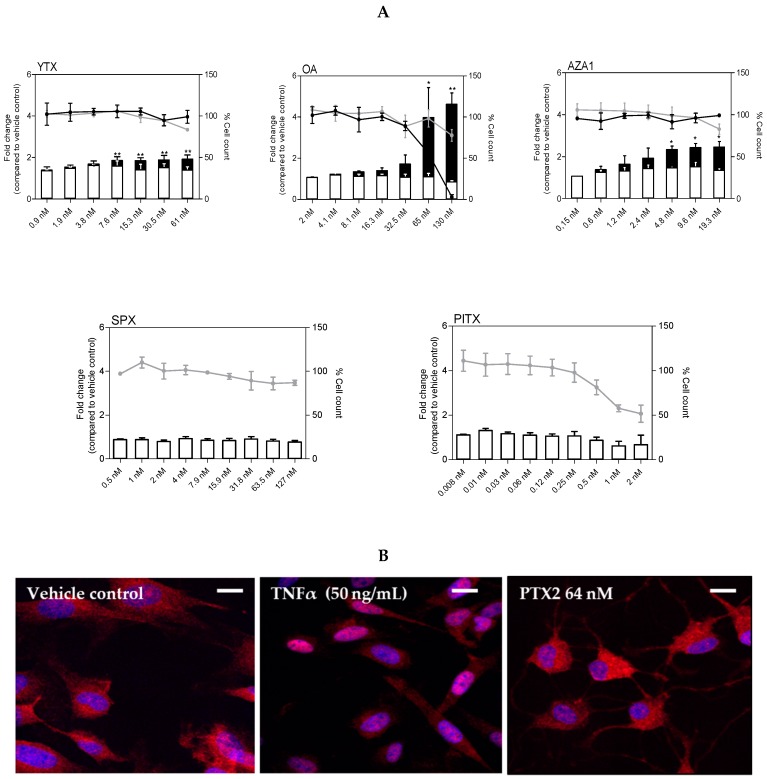
NF-κB nuclear translocation in EGCs after exposure to PTX2, YTX, OA, AZA1, SPX, and PlTX. (**A**) NF-κB-p65 nuclear translocation after 3 h (white) or 8 h (black) exposure was carried out by HCA. DAPI staining was used for cell count. NF-κB-p65 nuclear translocation was expressed as fold change compared to the vehicle control set to 1. Cell count values after 3 h (grey line) or 8 h (black line) exposure are expressed as percentages of the vehicle control. Values are presented as means ± SEM. Three independent experiments were performed. *, **: values significantly different from the vehicle control (respectively *P* < 0.05 and *P* < 0.01). (**B**) NF-κB-p65 (red) and nucleus (blue) after 3 h exposure to TNFα (50 ng/mL) and PTX2 (64 nM) were observed by confocal imaging. Scale bar = 20 µm.

**Figure 7 marinedrugs-17-00429-f007:**
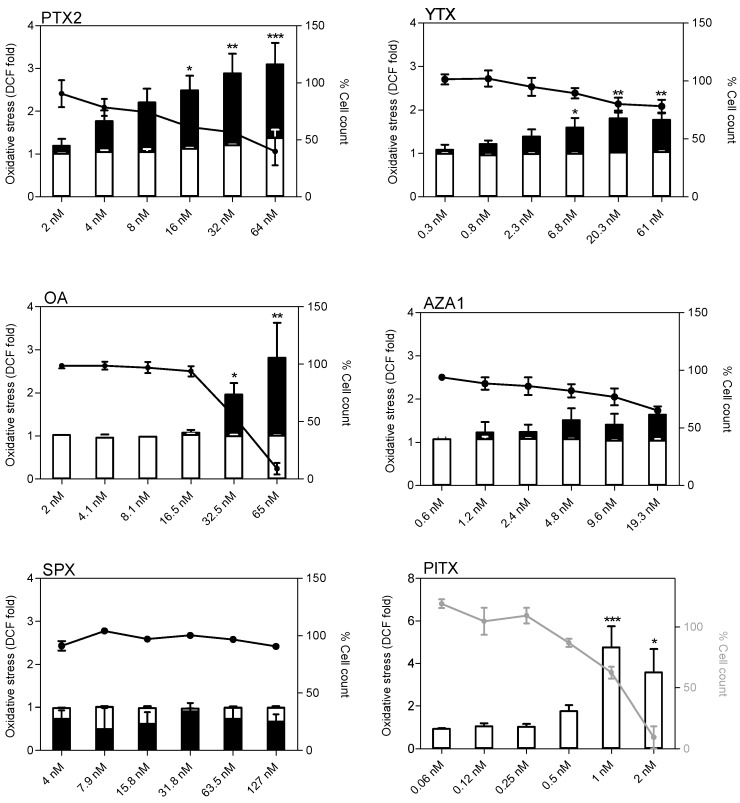
Oxidative stress in EGCs after exposure to PTX2, YTX, OA, AZA1, SPX, and PlTX. Measurement of the fluorescent DCF product after 4 h (white) or 24 h (black) exposure was carried out by HCA. Hoescht staining was used for cell count. ROS production was expressed as fold change compared to the vehicle control set to 1. Cell count values after 4 h (grey line) or 24 h (black line) exposure are expressed as percentages of the vehicle control. Values are presented as mean ± SEM. Four independent experiments were performed. *, **, ***: values significantly different from the vehicle control (respectively *P* < 0.05, *P* < 0.01, and *P* < 0.001).

**Figure 8 marinedrugs-17-00429-f008:**
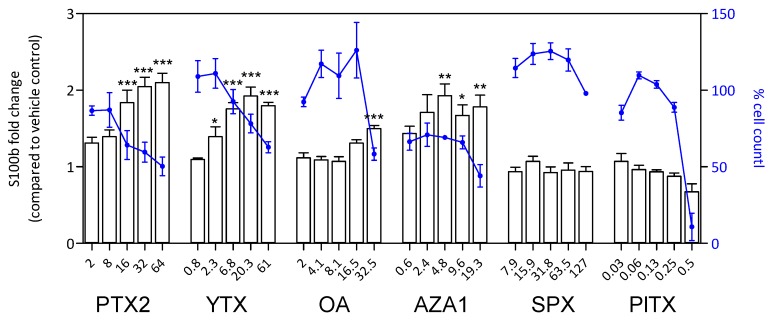
S100β response in EGCs after 24 h of phycotoxin exposure (nM). The measurement of S100β was carried out by HCA. DAPI staining was used for cell count (blue). S100β was expressed as fold change compared to the vehicle control set to 1. Cell count values are expressed as percentages of the vehicle control. Values are presented as mean ± SEM. Three independent experiments were performed. *, **, ***: significantly different from the vehicle control (respectively *P* < 0.05, *P* < 0.01, and *P* < 0.001).

**Figure 9 marinedrugs-17-00429-f009:**
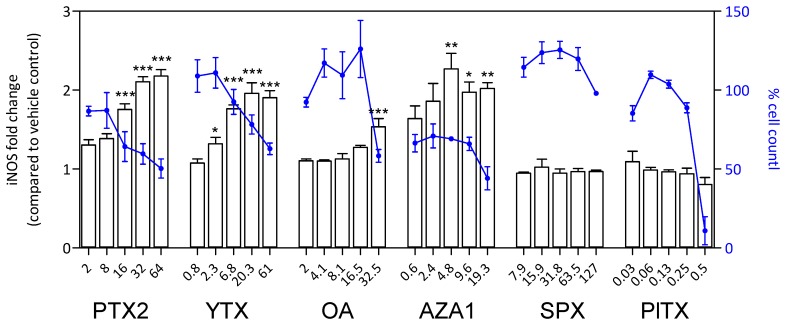
iNOS response in EGCs after 24 h of phycotoxin exposure (nM). The measurement of iNOS was carried out by HCA. DAPI staining was used for cell count (blue). iNOS was expressed as fold change compared to the vehicle control set to 1. Cell count values are expressed as percentages of the vehicle control. Values are presented as mean ± SEM. Three independent experiments were performed. *, **, ***: significantly different from the vehicle control (respectively *P* < 0.05, *P* < 0.01, and *P* < 0.001).

**Figure 10 marinedrugs-17-00429-f010:**
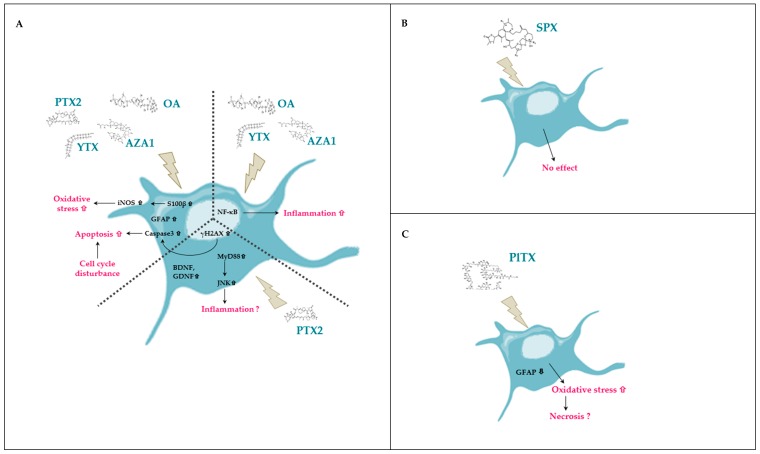
Schematic representation of the effects of PTX2, YTX, OA, AZA1, SPX, and PlTX induced on the rat EGC cell line CRL2690 illustrating the common and unique responses of each toxin. (**A**) Common effects and unique effects of PTX2, YTX, OA, and AZA1 on EGCs. (**B**) Effects of SPX on EGCs. (**C**) Effects of PlTX on EGCs.

**Table 1 marinedrugs-17-00429-t001:**
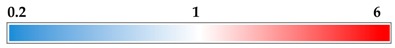
Gene expression in EGCs after 24 h exposure to PTX2, YTX, and AZA1. The analysis of gene expression was carried out by RT-qPCR. The resulting fold changes were calculated normalized to the reference gene GAPDH and the solvent-treated control. The heatmap shows the relative fold change values (red: up regulated and blue: down regulated) as well as statistical analysis of three independent experiments. *, **, ***: significantly different from the vehicle control (respectively *P* < 0.05, *P* < 0.01, and *P* < 0.001). ^a^: significant concentration effect (*P* < 0.05).

Biological Function		PTX2 (nM)		YTX (nM)		AZA1 (nM)
Gene	1	2	4		1	2	4		0.38	0.75	1.5
**Viability**	*BNIP3*						**					*
*CASP3*			***								
*GABARAP*											*
*FOS*						*	*			*	*
*BCL2*							*				
**Morphology**	*GFAP*	^a^	^a^	^a^								
**Cell cycle**	*CDK1*					*	***	***			*	**
*CDK2*						*	**			**	**
*CCNA2*					**	***	***				*
**Inflammation**	*IL1R1*							**				
*MAPK8*		*							*		
*CCL2*						*				*	*
*MYD88*			**								
*RHOA*							*				
*TLR4*											*
*FZD4*							*				
**Oxidative stress**	*CAT*											
*NFE2L2*											**
**Gliomediator**	*BDNF*											
*GDNF*							*				
**Channel and receptor**	*GJA1*							**				
*GFRA1*		*			*		*		*	*	*
*LPA1*			*								

**Table 2 marinedrugs-17-00429-t002:** Summary of in vitro toxicity of six phycotoxins on the rat EGC cell line CRL2690. Cell viability, morphological cell changes, oxidative stress, inflammation, cell cycle, and gliomediator expression were evaluated using RT-qPCR and high content analysis (HCA) approaches. n.e.: no effect. +, ++, +++: low, moderate, and high effect, respectively. n.t.: not tested.

Toxins	MORPHOLOGY		VIABILITY			CELL CYCLE	OXIDATIVE STRESS	INFLAMMATION	GLIOMEDIATORS		
	Microscopy	GFAP	IC_50_ (nM)	Caspase-3	γH2AX	Phases	Oxidative marker	NF-κB	S100β	iNOS	Gliomediator genes
PTX2	neurites alteration	++	n.e.	++	++	subG_1_ ↗ G_2_/M ↗	+++ (24 h)	n.e. (3 h)	++	++	BDNF ↗ GDNF ↗
YTX	neurites alteration	++	14.5	++	n.e.	subG_1_ ↗ G_2_/M ↘	++ (24 h)	+ (8 h)	++	++	GDNF ↗
OA	cell rounding	+	75.9	+	n.e.	subG_1_ ↗	+ (24 h)	++ (8 h)	+	+	n.t.
AZA1	neurites alteration	++	7.0	++	n.e.	subG_1_ ↗ G_2_/M ↘	+ (24 h)	++ (8 h)	++	++	n.e.
SPX	n.e	n.e.	n.e.	n.e.	n.e.	n.e.	n.e.	n.e. (3 h)	n.e.	n.e.	n.t.
PlTX	blebbing	-	0.4	n.e.	n.e.	n.e.	++ (4 h)	n.e. (3 h)	n.e.	n.e.	n.t.

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
