# Peer review of "Novel Insights on the Toxicity of Phycotoxins on the Gut through the Targeting of Enteric Glial Cells"

_marinedrugs, 2019, doi:10.3390/md17070429_

Reviewer 1 Report

The authors presented a very extensive study of the effects of six marine toxins on a rat EGC cell line. This is an in-depth work, and should be acceptable provided the followings are revised.

1) please quantify Fig2 results on cell shape changes

2) please shorten the Discussion, make it more concise

3) should provide a concise table to summarize the effects of the six toxins

Author Response

Dear Reviewer,

We would like to thank you for your consideration about our submitted publication and the quality of the comments. We got all complementary information that was requested, and greatly improved the manuscript. Please find below your comments followed up by our responses :

Please quantify Fig2 results on cell shape changes

Response: Fig2 is just a qualitative observation on cell shape change. The Fig3, particularly data on cell body area, quantifies the morphological change. We have tried to have a data about quantification of the neurites change (length, number,..) but the GFAP staining was very low

Please shorten the Discussion, make it more concise

        Response: The discussion has been reduced

   3. Should provide a concise table to summarize the effects of the six toxins

        Response: The Table2 has been changed

Thank you for your consideration,

Sincerely,

Reviewer 2 Report

Lipophilic shellfish toxins including PTX2,YTX, OA, AZA1, SPX, PlTX cause significant economic hardship for shellfish industries worldwide, in terms of monitoring costs and farm closures, even though no human fatalities have ever been associated with them. The major impacts on human health are intestinal problems such as diarrhea and stomach cramps, notably from OA and AZA1, but human health significance from oral ingestion of YTX, PTX, PlTX analogs, SPX remains debated, and regulatory practices are in state of flux. The present work represents a massive, competently executed, experimental study challenging above pure toxin fractions against enteric glial cells using the CRL2690 rat cell line. Has this cell line been well studied previously, and how do the results compare with the more widely studied Caco-2 human intestinal cell line?The work focuses on cell line viability, morphology, and gene expressions reflecting cell cycle, inflammation, oxidative stress, and several specific glial markers such as GFAP etc. The results are significant and deserve to be widely read BUT this is a very long manuscript (27 pages) full of acronyms (ECGs, ENs, RNU, iNOS) which is hard work to digest. Are all the figures needed? Could this manuscript perhaps be cut in two and eg the gene expression and HCA approaches presented separately? I recommend that the Tables1 and 2 should be supplementary material. The  discussion abruptly ends with Table 3 that summarises the results, as well as a schematic representation in Fig.9, but these are not well described. Please  try and explain this work better to a non-expert but interested general reader.

Some minor details:

title + throughout: insights (plural)

Line 12. cells underlying the epithelium

L38. depicted=described from

L70.reported=demonstrated

L92. throughout. "vehicle"=? (carrier) medium, ? control medium

L120; L162. depicted=observed

L267. what are clone 9 cells?

L279. deteriorated=adversely affected

L288, L294, L318. Avoid using "in fact"

L301. increase in genes=?gene expression

L330. linked to

L424. what means "passages 38-58"?

L471. Cell counts were performed

L525. from PTX2

Author Response

Dear Reviewer,

We would like to thank you for your consideration about our submitted publication and the quality of the comments. We got all complementary information that was requested, and greatly improved the manuscript. Please find below your comments followed up by our responses

The results are significant and deserve to be widely read BUT this is a very long manuscript (27 pages) full of acronyms (ECGs, ENs, RNU, iNOS) which is hard work to digest. Are all the figures needed? Could this manuscript perhaps be cut in two and eg the gene expression and HCA approaches presented separately? I recommend that the Tables1 and 2 should be supplementary material. The discussion abruptly ends with Table 3 that summarises the results, as well as a schematic representation in Fig.9, but these are not well described. Please try and explain this work better to a non-expert but interested general reader.

Response: Although data are significant, one publication seems to be more relevant than 2 publications because all data are complementary using different approaches. Moreover, we have tried to use fewer acronyms as you have suggested (especially ENS). Table1 have been changed to Fig4, we hope that it is clearer. Table2 has been changed to a new table which facilitates the understanding. Fig10 (Fig9 in the previous version) has been more described at the end of the discussion. The discussion has been reduced and we hope that this last reviewed version will be more concise as you advised us. We took into account your comments and we have changed the words that you have underlined (from L12 to L525).

Thank you for your consideration,

Sincerely